# Cluster Analysis of Home Polygraphic Recordings in Symptomatic Habitually-Snoring Children: A Precision Medicine Perspective

**DOI:** 10.3390/jcm11195960

**Published:** 2022-10-09

**Authors:** Marco Zaffanello, Angelo Pietrobelli, David Gozal, Luana Nosetti, Stefania La Grutta, Giovanna Cilluffo, Giuliana Ferrante, Michele Piazza, Giorgio Piacentini

**Affiliations:** 1Department of Surgical Sciences, Dentistry, Gynecology and Pediatrics, University of Verona, 37129 Verona, Italy; 2Departments of Child Health, and Medical Pharmacology and Physiology, School of Medicine, University of Missouri, Columbia, MO 65212, USA; 3Pediatric Sleep Disorders Center, Division of Pediatrics, F. Del Ponte Hospital, Insubria University, 21100 Varese, Italy; 4Institute of Translational Pharmacology (IFT), National Research Council, 90146 Palermo, Italy; 5Department of Earth and Marine Sciences, University of Palermo, 90133 Palermo, Italy

**Keywords:** children, cluster analysis, obstructive sleep apnea, polygraphy, sleep-disordered breathing, sleep apnea, snoring

## Abstract

(1) Background: Sleep-disordered breathing (SDB) is a frequent problem in children. Cluster analyses offer the possibility of identifying homogeneous groups within a large clinical database. The application of cluster analysis to anthropometric and polysomnographic measures in snoring children would enable the detection of distinctive clinically-relevant phenotypes; (2) Methods: We retrospectively collected the results of nocturnal home-based cardiorespiratory polygraphic recordings and anthropometric measurements in 326 habitually-snoring otherwise healthy children. K-medoids clustering was applied to standardized respiratory and anthropometric measures, followed by Silhouette-based statistics. Respiratory Disturbance Index (RDI) and oxygen desaturation index (≤3%) were included in determining the optimal number of clusters; (3) Results: Mean age of subjects was 8.1 ± 4.1 years, and 57% were males. Cluster analyses uncovered an optimal number of three clusters. Cluster 1 comprised 59.5% of the cohort (mean age 8.69 ± 4.14 years) with a mean RDI of 3.71 ± 3.23 events/hour of estimated sleep (e/ehSleep). Cluster 2 included 28.5% of the children (mean age 6.92 ± 3.43 years) with an RDI of 6.38 ± 3.92 e/ehSleep. Cluster 3 included 12% of the cohort (mean age 7.58 ± 4.73 years) with a mean RDI of 25.5 ± 19.4 e/ehSleep. Weight z-score was significantly lower in cluster 3 [−0.14 ± 1.65] than in cluster 2 [0.86 ± 1.78; *p* = 0.015] and cluster 1 [1.04 ± 1.78; *p* = 0.002]. Similar findings emerged for BMI z scores. However, the height z-score was not significantly different among the 3 clusters; (4) Conclusions: Cluster analysis of children who are symptomatic habitual snorers and are referred for clinical polygraphic evaluation identified three major clusters that differed in age, RDI, and anthropometric measures. An increased number of children in the cluster with the highest RDI had reduced body weight. We propose that the implementation of these approaches to a multicenter-derived database of home-based polygraphic recordings may enable the delineation of objective unbiased severity categories of pediatric SDB. Our findings could be useful for clinical implementation, formulation of therapeutic decision guidelines, clinical management, prevision of complications, and long-term follow-up.

## 1. Introduction

Sleep-disordered breathing (SDB) is a common problem in children. Indeed, the American Academy of Pediatrics (AAP) Subcommittee on Pediatric Sleep estimated that 1.2% to 5.7% of children might suffer from obstructive sleep apnea (OSA) [1], with the peak incidence occurring between two and six years of age. Adenotonsillar hypertrophy and obesity are major risk factors for OSA in otherwise healthy children [2,3]. Overnight in-lab polysomnography is currently the gold standard for diagnosing OSA in children. However, home cardio-respiratory polygraphy (HRP) has emerged as a potentially useful and reliable approach [4].

The current classification of SDB is based on the severity of the symptoms as well as its pathophysiology, although the dominant element in the attribution of severity categories relies on the number of respiratory events per hour of sleep, i.e., the Respiratory Disturbance Index (RDI) or AHI (Apnea-Hypopnea Index). This classification has demonstrated its value in predicting the risk of SDB-associated morbidities [5,6,7], but it is grossly inaccurate in effectively defining phenotypes that may guide a more personalized clinical management. Clinical experience has also identified factors such as somatic growth (i.e., overweight-obese, normal weight, underweight) and pathophysiologic elements (i.e., lymphatic hyperplasia, obesity, craniofacial) as important modifiers of the clinical presentation, the response to treatment and outcomes in adulthood [8], further illustrating the need for objective and quantifiable approaches to the classification of clinical SDB traits.

Person-centered statistical methods, such as cluster analysis, offer a unique opportunity to identify singular homogeneous groups that are contained within any given cohort of participants in a clinical study. Clustering methods aim to minimize the sum of variances within the cluster and to maximize the separation of clusters, thus allowing for the identification of distinct phenotypic groups within any given cohort [9]. To the best of our knowledge, very few studies have applied cluster analysis to phenotype SDB in children, while multiple reports have been published among adults. These studies primarily focused on characterizing clinical aspects and did not incorporate objective sleep measures, such as those that are routinely collected during HRP. Wu et al. reported three homogeneous groups in children with SDB when evaluating questionnaire-based symptoms along with clinical comorbidities. Specifically, one of the identified clusters was characterized by snoring and daytime sleepiness, whereas symptoms of daytime hyperactivity and minor symptoms characterized the other two clusters [10]. In adults with OSA, clinical variables have been grouped into different clusters and have included PSG results [11,12,13,14,15].

In our cross-sectional study, we hypothesized that the cluster analysis could identify the number and characteristics of clusters within a large dataset of habitually snoring and otherwise healthy children who had been referred at the first outpatient assessment. The data of interest for this analysis were: the number and type of nocturnal respiratory events measured with home HRP and body growth.

## 2. Materials and Methods

### 2.1. Study Design and Population

This retrospective-cross-sectional study was conducted following the Declaration of Helsinki and under the terms of local legislation. The protocol was approved by the Ethics Committee for clinical trials in the provinces of Verona and Rovigo at Integrated University Hospital (CESC601). All parents/caregivers gave informed consent for the scientific use of the data.

The HRP results of 470 children who were clinically assessed between the years 2013 and 2015 at the pediatric outpatient clinic (Verona, Italy) for SDB were reviewed and incorporated into a database. Figure 1 shows the flowchart of the inclusion and exclusion criteria applied for children included in the initial cohort. The exclusion criteria were: HRP recordings lasting less than 7 h, age < 2 years, absence of follow-up medical records, presence of genetic syndromes or of malabsorption syndromes (e.g., celiac disease), genetic disorders (Prader–Willi, Down syndrome, congenital adrenal hyperplasia, craniofacial disorders), other known chronic conditions (e.g., neurological and/or muscular disorders) and any other types of sleep disorders, such as insomnia, parasomnia, and delayed sleep phase syndrome. A total of 144 charts were excluded because HRP recordings <7 h (*n* = 25) and/or age < 2 years (*n* = 65) and/or genetic disorders (*n* = 30) and/or neurological diseases (*n* = 9) and duplicate follow-up records (*n* = 53). The total number of unique identifiers subjects included in the final analysis was 326 habitually-snoring children.

### 2.2. Anthropometry

Height and weight were measured using a mechanical medical precision scale (WUNDER C201, Wunder Sa.Bi. srl, Milan, Italy) along with a telescopic stadiometer. Specially trained health personnel measured children wearing light clothing and no shoes. Weight was recorded to the nearest 0.1 kg, and height was measured to the nearest 0.1 cm.

Underweight was defined as less than the 5th percentile for age and height, overweight as 85th–<95th percentile, and obesity was defined as ≥95th percentile. Body mass index [weight (kg)/height (m^2^)], BMI percentiles, and BMI z-scores were calculated using an online tool (http://www.bcm.edu/bodycomplab/BMIapp/BMI-calculator-kids.html, accessed on 10 June 2022), which is based on CDC growth charts for children and teens, ages 2 through 19 years: (https://www.cdc.gov/healthyweight/bmi/calculator.html, accessed on 10 June 2022).

### 2.3. Pediatric Home Respiratory Polygraphy

Individual HRP physiological recordings were stored in the designated database, including the analysis software. Of note, the HRP database is preserved for at least 10 years. The overnight HRP studies were performed using a portable outpatient device (SOMNOscreenTM PSG, SOMNOmedics GmbH, Randersacker, Germany) with continuous nasal airflow monitoring by cannula, thoracic and abdominal breathing movements (chest and abdominal belts), arterial oxygen saturation (SpO_2_; digital pulse oximetry), heart rate and electrocardiogram trace (ECG), body position (mercury sensor) and tracheal sounds (microphone), as previously described [16]. The unit has been programmed to turn on automatically and turn off automatically, depending on the child’s sleep habits. Parents were trained to use the device correctly and recorded in a diary the time when the child fell asleep, any nocturnal awakenings, and the time of awakening in the morning.

Analysis of the recordings was initially performed by the software (DOMINO software, Somnomedics v.2.6.0, Am Sonnenstuhl 63, D-97236 Randersacker) and then carefully checked by one of the investigators (MZ). The estimated Total Sleep Time [eTST (hours)] was calculated according to the published criteria [17,18] and served as the denominator for any index calculation. Respiratory events were evaluated according to the American Academy of Sleep Medicine guidelines [17]. The total number of obstructive apnea (OA), hypopnea (H), mixed apnea (MA), and central apnea (CA) was then divided by the duration of eTST (hour) to derive the index of each type of respiratory event (n./h by eTST). The Oxygen Desaturation Index (ODI; n./hour) was calculated as the number of episodes of O_2_ desaturation ≥ 3% below baseline per hour of eTST (ODI3%). Average SpO_2_ and minimum SpO_2_ were also automatically calculated. Snoring events (expressed as % of eTST) in the overall nocturnal recording were also computed [17]. Other parameters evaluated in the study were the fraction of time and corresponding respiratory indices that occurred in the supine and non-supine sleeping positions. Since the consensus classification of the severity of SDB in children is routinely based on the RDI, an obstructive RDI ≤ 1/hour of eTST was defined as normal, 1 < RDI ≤ 5/hour of eTST was defined as mild OSA, 5 < RDI ≤ 10 /hour of eTST was defined as moderate OSA, and an RDI >10/hour of eTST was defined as severe OSA [19].

### 2.4. Statistical Analysis

Data are presented as *n* (%) or mean (SD). Differences in categorical variables were analyzed using the Chi-square test, and quantitative variables were compared using the Kruskal–Wallis test.

K-medoids clustering was applied using standardized RDI, supine OA, not supine OA, supine CA, not supine CA, supine MA, not supine MA, and ODI. Given a set of variables (x1, x2, …, xm), clustering of k-medoids aims to partition the *n* observations into k (≤*n*) sets S = {S1, S2, …, Sk} to minimize variances within the cluster by applying the following equation:$$\underset{S}{\arg\;\min}\overset{k}{\underset{i=1}{\sum }}\underset{x\in S_{i}}{\sum }|\left|x-\mu_{i\;}\right|{|}^{2}$$

$\mu_{where\;}$ are the centroids in $S_{i}$. The algorithm starts with the first group of randomly selected centroids, which are used as starting points for each cluster, and then performs iterative calculations to optimize the positions of the centroids. The algorithm stops if there are no changes in the centroid, thereby indicating that the clustering function was successful after a maximum number of iterations was reached. We used partitioning around the Medoid algorithm since it is more robust to outliers and noise than the K-Means algorithm because it minimizes the sum of the dissimilarities between the points labeled into any given cluster and also provides a point designated as the center of that cluster (Medoids), instead of using the midpoint [20]. Silhouette statistics were used to determine the optimal number of clusters [21]. The Benjamini Hochberg method was used to adjust *p*-values for multiple comparisons. All statistical analyses were performed using version R 4.0.2 (R Foundation for Statistical Computing, Vienna, Austria). Statistical significance was set at a two-tailed *p* < 0.05.

## 3. Results

The summary of the relevant clinical data corresponding to 326 children included in the final analysis is shown in Table 1, Panel A. Subjects had a mean age of 8.1 years, and 57% were male. According to the BMI z-score, 5.2% of the children were underweight, and 25.8% were obese. The summary of HRP results of the 326 children is shown in Table 1, Panel B. Their mean RDI was 7.08 ± 10.10 /hour of eTST, and the mean ODI3% was 4.54± 9.22 /hour of eTST. According to the RDI severity criteria, 24.2% and 17.2% of the enrolled children had mild and severe OSA, respectively.

A comparable result of average silhouette width was observed for three, six, and eight clusters (Figure 2). The optimal number of clusters k is the one that maximizes the average silhouette over a range of possible values for k. However, according to the best trade-off between silhouette method, cluster size, and interpretability, the optimal number of clusters chosen was three.

The characterization of the clusters is shown in Table 2 (panels A and B). Cluster 1 included 194 children [59.5% of the total; 112 (57.7%) males] with a mean age 8.69 ± 4.14 years. This group showed a mean RDI value of 3.71 ± 3.23 /hour of eTST. Cluster 2 included 93 children [28.5% of the total; 54 (58,1%) males] and the mean age was 6.92 (3.43) years, significantly younger than cluster 1 [*p* < 0.005]. The mean RDI for cluster 2 was of 6.38 ± 3.92 /hour of eTST. Cluster 3 included 39 children [12% of the total; 20 (51%) males], and their mean age was 7.58 ± 4.73 years, while their mean RDI was 25.5 ± 19.4 /hour of eTST.

OA was more frequent in the supine position, and the percentage of time spent in the supine position during sleep was comparable across the three clusters (*p* = 0.567) (Table 2, Panel B). The mean supine OA differed significantly between Cluster 3 [17.0 ± 21.8/hour of eTST] and cluster 1 [2.19 ± 3.28/hour of eTST; *p* < 0.001] and between Clusters 3 and 2 [3.07 ± 5.47/hour of eTST; *p* < 0.001]. The mean percentages of OA observed in the supine position were: 64.4% in Cluster 1, 67.3% in Cluster 2, and 66.6% in Cluster 3 (*p*—not significant).

The mean indices of CA were significantly different between clusters. The mean CA was higher in Cluster 2 than in Cluster 1 [3.53 ± 1.91/hour of eTST vs. 0.91 ± 0.71/hour of eTST, respectively; *p* < 0.001] and Cluster 3 [2.39 ± 2.74/hour of eTST; *p* < 0.001]. The index of CA was higher in Cluster 2 [4.37 ± 7.96 /hour of eTST] in the supine position than in Cluster 1 [0.98 ± 1.09/h of eTST; *p* < 0.001] and Cluster 3 [3.63 ± 9.61/hour of eTST; *p* = 0.016]. In contrast, the mean index of CA in the supine position did not differ between Clusters 1 and 3 (*p* = 0.757).

Table 2, Panel A shows additional HRP-related findings. In particular, the mean SpO_2_ (expressed as % of eTST) was lower in Cluster 3 [94.7 ± 6.50%] than in Cluster 1 [97.1 ± 1.08%; *p* < 0.001] and Cluster 2 [97.1 ± 1.79%; *p* < 0.001], the latter two being similar. The mean time spent with SpO_2_ < 90% was greater in Cluster 3 [5.95 ± 15.6%] than in the other two clusters [Cluster 1: 0.48 ± 2.29% and Cluster 2: 1.02 ± 4.61%; *p* < 0.001, respectively]. The mean time spent with SpO_2_ < 90% was comparable between Clusters 1 and 2 (*p* = 0.768).

Table 2, panel B, and Figure 3 show the comparison of the weight z-scores (Chart A), height (SD) z-score (Chart B), and BMI (SD) z-score (Chart C) of the three clusters. The weight z-score was significantly lower in Cluster 3 [−0.14 ± 1.65] compared to Cluster 2 [0.86 ± 1.78; *p* = 0.015] and cluster 1 [1.04 ± 1.78, *p* = 0.002]. No statistical differences emerged for weight z-scores between Clusters 1 and 2. Height z-scores were not significantly different among clusters (*p* = NS). Consequently, BMI z-scores were significantly higher in cluster 1 [1.50 ± 1.91] compared to cluster 2 [0.97 ± 1.72; *p* = 0.044], but not compared to Cluster 3 [0.86 ± 2.42; *p* = 0.63].

BMI categories (underweight, normal weight, overweight and obese) were statistically different among clusters (*p* = 0.015; Table 2). Specifically, BMI categories were significantly different between Clusters 3 and 1 (*p* = 0.033). The percentage of underweight children increased progressively from clusters 1 to 3 (2.58%, 6.45%, and 15.4%, respectively). Moreover, cluster 3 had fewer overweight children (12.8%) than cluster 1 (24.8%) and cluster 2 (21.5%). Similarly, cluster 1 had more overweight children than cluster 3 (+11.4%). The percentages of children with weight, height, and BMI < 5th percentile were not statistically different between clusters.

## 4. Discussion

In the present study, the application of cluster analysis using HRP findings and anthropometric measurements allowed for the identification and characterization of three different phenotypes of childhood OSA. Each cluster exhibits different mean RDI values and shows differences in mean weight z-score and mean BMI z-score. When compared to the current standard definitions of severity, each of the clusters resides within the established RDI -derived categories of mild, moderate, and severe, respectively.

Statistical power was assessed using the R package clusterPower (Kleinman K, Sakrejda A, Moyer J, Nugent J, Reich N, Obeng D. Power Estimation for Randomized Controlled Trials: clusterPower; Version 0.7.0 (2021). https://cran.r-project.org/package=clusterPower, accessed on 16 September 2022). Considering three clusters and the distribution of OSAS severity on 326 subjects, statistical power of 72% was obtained.

The present study showed that children who come to the first outpatient evaluation because they snore and have a growth deficit should be suspected of having severe SDB. Conversely, overweight children may be allocated among those with mild-moderate SDB. By considering symptoms and growth parameters, doctors may suspect the severity of SDB in children who come to the first outpatient evaluation. To the best of our knowledge, there are only two studies: one in children and another in adult patients, in which cluster analysis was applied to investigate the relationship between SDB and body growth [22,23]. Freeman and Bonuck evaluated 10,441 children in whom the severity of SDB was based on a longitudinally repeated questionnaire (at ages 6, 18, 30, 42, 57, 69, and 81 months), and categorized as “snoring”, “apnea” and “mouth breathing”. The authors identified five different phenotypes. The “early snorers” cluster was significantly shorter than the “normals” cluster (*p* = 0.0038). Children categorized as “normals” showed the lowest BMI, and the “early snorers” group had the highest BMI (*p* < 0.05) [22]. In the study by Nakayama et al., three clusters based on supine PSG results were identified among 210 adult patients with moderate and severe OSA with differences in BMI and type of apnea. The three clusters included: (1) obese patients with high apnea fraction (ratio of apneas to a total number of respiratory events) and severe desaturations; (2) non-obese patients with high apnea fraction and long duration of the respiratory event and (3) patients with low apnea fraction and high central apnea percentage [23].

Spruyt et al. performed a hierarchical cluster analysis of PSG studies along with other demographic and anthropometric data in 1,133 children (aged 5 to 9-year-old) who were originally recruited from the community [24]. This cohort was enriched for habitual snorers (52.8% were habitual snorers). The investigators found six different phenotypic clusters. BMI within cluster 5 (AHI 18.1 ± 11.6; BMI 21.5 ± 5.8) was higher than in clusters 1 (AHI 0.8 ± 0.9; BMI 18.0 ± 4.5) and cluster 4 (AHI 1.3 ± 2.2; BMI 17.3 ± 4.3) [24]. These authors showed that children with elevated BMI were more likely to suffer from severe OSAS, unlike our current study. It is possible that the different demographic and racial characteristics, as well as the enrolment strategy (community and asymptomatic vs. clinically symptomatic referral children), may account for the differences, further stressing the importance of regional phenotypes in the context of a better personalization of any methodological approach aimed at improved clinical precision.

Previous studies have already reported that OSA severity and frequency are generally worse when patients are in the supine position [25,26,27]. The term positional OSA (POSA) is applied when the obstructive AHI is ≥2-fold more severe in the supine position [26,27]. In our study, this ratio was 1.81 (1.89 for OA + H) in cluster 1, 2.06 (1.92 for OA + H) and in cluster 3 was 1.75 (1.59 for OA + H). Some studies have shown that POSA is associated with obesity in children. Dayyat et al. showed that OSA worsens in the supine position in obese children. Indeed, the AHI was significantly higher in the supine position than in either the lateral or prone positions. The authors suggested that these children were more likely to be sleeping prone because this position can facilitate the patency of the upper airway [28]. Similarly, Tholen and colleagues reported that the supine position worsens AHI in obese children when compared to the non-supine position) [29]. In this report, 58% of the 112 children with obesity had POSA, and of these, 52% had mild OSA, 28% had moderate OSA, and 20% had severe OSA. Thus, among children with obesity and OAs, POSA occurs frequently [30].

In cluster 3, the number of OA was very high compared to clusters 1 and 2. However, the number of OA (n./h) and H (n./h) were not statistically different between clusters 1 and 2. The increased number of CAs in cluster 2 compared to cluster 1 accounted for the differences in AHI means between the two clusters. The increased presence of central events was the hallmark of cluster 2, along with the allocated children being younger and thinner. As far as we know, there are no references in the literature. Sleep CAs have been attributed to the instability of the feedback mechanisms that control breathing. Certain medical conditions, the most common of which are neuroanatomical abnormalities, can cause a significant increase in CAs or CSA. However, although the prevalence is higher in preschool-aged children, CSA is relatively rare in otherwise school-aged healthy children, and our demographics were insufficient to explain this finding. Some studies have reported that CAs are not associated with major desaturations in healthy children [31]. In children with mild OSA, the presence of a] respiratory comorbidity such as asthma is associated with more CAs than in children without this condition. It has been hypothesized that the presence of wheezing or asthma may worsen systemic inflammation, and the latter may alter peripheral and central chemoreceptive mechanisms [32]. Furthermore, in patients with OSA, which is accompanied by intermittent hypoxia and fragmented sleep, increased generation of free radicals and inflammation will occur [33]. These phenomena may yield alterations of the central and peripheral nervous system and consequent dysfunction [34]. Thus, it is posited that the coexistence of recurrent upper airway collapse and respiratory control instability may trigger episodes of CA [35,36]. As illustrated by our cluster constructs, the mean values of CA and ODI were higher in cluster 2 and cluster 3. This was an unexpected finding since CA would be particularly anticipated in the cluster with the more prominent alterations in oximetry parameters. Furthermore, in the supine position, the mean of CA was higher in cluster 2 and comparable in clusters 1 and 3, although a substantial variance was present in cluster 3. We should point out that the increase in CA in children with OSAS was reported by Boundewyns et al., who found that in about two-thirds of children [mean age 2.7 (1.5–4.6) years] suffering from OSA [oAHI 10.6 (5.2–21.7) events/hour] there was evidence of CA [2.4 (1.7–3.8) events/hour]. Furthermore, these investigators found a significant correlation between CAI and minimal SpO_2_, but not with BMI-z score [37]. In contrast, we found that cluster 3 with more severe OSAS had higher mean MA values than clusters 1 and 2. MA usually develops when a brief period of CA is followed by an OA. We can therefore speculate that children included in cluster 3 may suffer from increased respiratory control instability (increased CA) which in turn may facilitate the occurrence of upper airway obstructive events, thereby manifesting as MA.

Overnight PSG is the gold standard for OSA diagnosis. Therefore, home-based HRP has clearly emerged as a less onerous alternative to in-laboratory PSG and is likely more representative of the child’s habitual nocturnal sleep [19]. HRP requires easy-to-use portable monitors in the child’s home, thereby leading to substantial relief in the natural hospital-induced child stress and apprehension [38,39]. Although Level 3 devices traditionally employed at least three channels (oximetry, airflow, and respiratory effort), we implemented HRP systems that included six channels, thereby providing greater detail and accuracy in the scoring analysis and interpretation of observed events. Obviously, another limitation of our study is the retrospective design and the absence of an interventional arm that may provide more causal relationships, the latter being precluded. It might be interesting to include the size of the tonsils in the cluster analysis. However, the study by Wu et al. showed that there was no statistical difference in the incidence of tonsil and adenoid hypertrophy between clusters, despite there being statistically significant differences in AHI and ODI [10]. For this reason, we decided not to include the dimensional classification of the tonsils in the cluster analysis. Furthermore, this being a cross-sectional study, we cannot estimate how long the subjects had altered respiratory features during sleep. It is possible that the longer the disease duration, the more pronounced the impact on somatic growth will be.

## 5. Conclusions

In summary, this study identified three main and distinct clusters among children with OSAS. Each cluster incorporated different anthropometric and respiratory components during sleep, which in fact, provide legitimacy to the previously arbitrary and subjective allocation of three OSA severity categories (mild, moderate, and severe), all the while using HRP as the diagnostic tool. However, the boundaries of each cluster may differ from one pediatric sleep center to the next, and as such multicenter prospective studies that incorporate much larger cohorts are needed. In addition, exploration of the potential cluster-dependency, and as a corollary, delineation of severity-dependent risk for end-organ morbidities, as well as prediction of treatment-associated outcomes within each cluster, may permit precision medicine for improved clinical management, prevision of complications, and long-term follow-up.

## Figures and Tables

**Figure 1 jcm-11-05960-f001:**
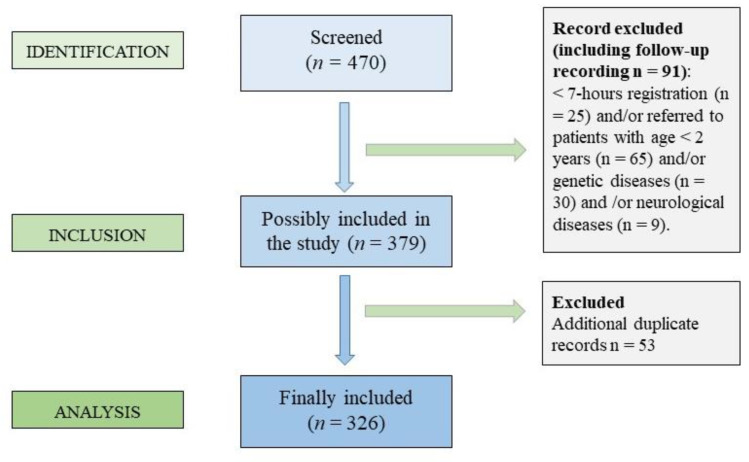
Flowchart showing the inclusion and exclusion criteria of the children who have performed the home cardio-respiratory polygraphy (HRP).

**Figure 2 jcm-11-05960-f002:**
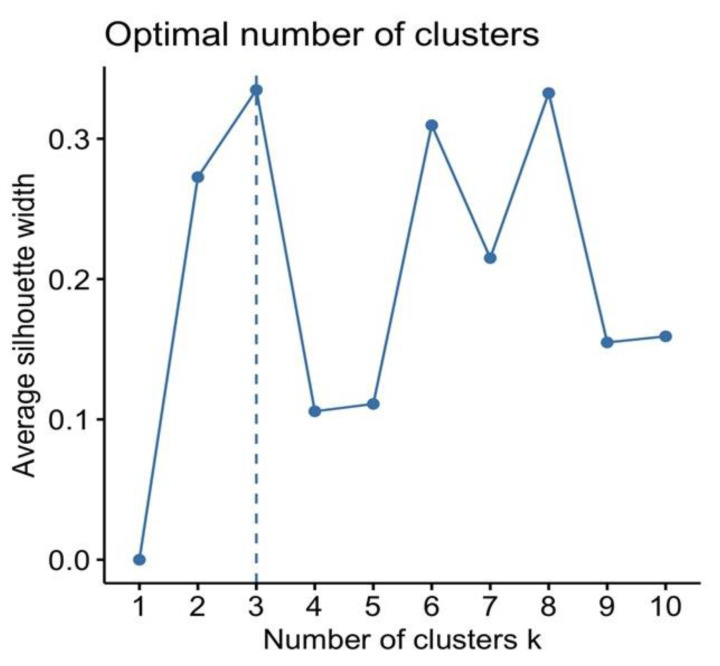
Optimal number of clusters according to the Silhouette method; the optimal number of clusters was three based on the highest average Silhouette value.

**Figure 3 jcm-11-05960-f003:**
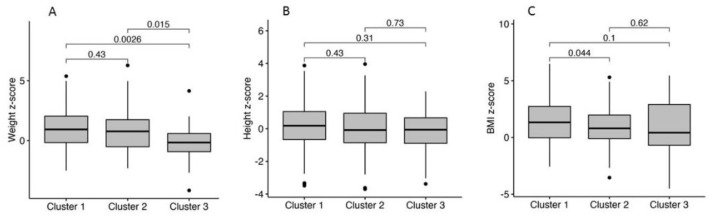
Comparison of normalized anthropometric parameters ((**A**) weight z-score, (**B**) height z-score, and (**C**) BMI z-score) among the three clusters.

**Table 1 jcm-11-05960-t001:** Characteristics of study population included in the final analysis (Panel A); Summary of HRP results of the study population (Panel B; *n* = 326).

Panel A			
		*n* = 326	
Age, years		8.05 (4.09)	
Sex: M, *n* (%)		186 (57.06)	
Height, cm		126.88 (25.12)	
Height, Percentile		55.06 (32.78)	
Height, Z-score		0.10 (1.39)	
Weight, kg		38.95 (28.59)	
Weight, Percentile		65.50 (34.67)	
Weight, Z-score		0.83 (1.79)	
BMI, kg/m^2^		21.14 (7.98)	
BMI, z-score		1.27 (1.94)	
BMI categories			
Underweight		17 (5.21)	
Normal weight		153 (46.93)	
Overweight		72 (22.09)	
Obese		84 (25.77)	
Panel B			
Respiratory Events	*n*./h (SD)	Event per Body Position	*n*./h (SD)
OA	3.09 (6.86)	Supine OA	4.21 (9.64)
MA	0.37 (1.19)	Not supine OA	2.30 (5.78)
CA	1.83 (1.89)	Supine CA	2.26 (5.65)
H	1.79 (3.91)	Not supine CA	1.85 (1.99)
RDI	7.08 (10.10)	Supine MA	0.49 (1.64)
ODI	4.54 (9.22)	Not supine MA	0.30 (0.98)
Minimum SpO_2_, mean (SD) %	86.19 (10.49)	OSAS severity	*n*. (%)
Mean SpO_2_, mean (SD) %	96.82 (2.68)	Mild	191 (58.59)
SpO_2_ < 90%, mean (SD) minutes	1.29 (6.37)	Moderate	79 (24.23)
Snoring, mean (SD) %TST	2.24 (5.73)	Severe	56 (17.18)

Abbreviations: BMI, Body Mass Index; M, males. CA, central apnea; h, hour; H, hypopnea; MA, mixed apnea; OA, obstructive apnea; ODI, oxygen desaturation index; OSAS, obstructive sleep apnea syndrome; RDI, respiratory disturbance index; SD, standard deviation; TST, total sleep time.

**Table 2 jcm-11-05960-t002:** Results of clusters characterization (n. 3) regarding the anthropometric parameters (panel A) and home polygraphic findings (panel B).

Panel A							
	Cluster 1	Cluster 2	Cluster 3	*p*-Value	Cluster 1 vs. Cluster 2	Cluster 1 vs. Cluster 3	Cluster 2 vs. Cluster 3
	*N* = 194	N = 93	N = 39				
Age, mean (SD) years	8.69 (4.14)	6.92 (3.43)	7.58 (4.73)	0.002	0.002	0.260	0.664
Sex: M, *n* (%)	112 (57.7%)	54 (58.1%)	20 (51.3%)	0.739	1.000	0.900	0.900
Weight, Percentile	69.5 (33.2)	60.3 (35.0)	57.8 (38.8)	0.036	0.088	0.128	0.919
Weight Z-score	1.04 (1.78)	0.86 (1.78)	−0.14 (1.65)	0.009	0.430	0.003	0.015
Height, Percentile	55.9 (32.4)	54.4 (33.8)	52.5 (32.7)	0.811	0.925	0.820	0.950
Height Z-score	0.17 (1.36)	0.03 (1.49)	−0.12 (1.31)	0.406	0.430	0.310	0.730
BMI, mean (SD) kg/m^2^	22.0 (8.14)	18.9 (5.36)	22.0 (11.0)	0.006	0.005	1.000	0.103
BMI z-score	1.50 (1.91)	0.97 (1.72)	0.86 (2.42)	0.034	0.044	0.100	0.620
BMI categories, *n*. (%)							
Underweight	5 (2.58)	6 (6.45)	6 (15.4)	0.015	0.259	0.033	0.259
Normal weight	87 (44.8)	48 (51.6)	18 (46.2)				
Overweight	47 (24.2)	20 (21.5)	5 (12.8)				
Obese	55 (28.4)	19 (20.4)	10 (25.6)				
Panel B							
Polygraphy	Cluster 1	Cluster 2	Cluster 3	*p*-Value	Cluster 1 vs. Cluster 2	Cluster 1 vs. Cluster 3	Cluster 2 vs. Cluster 3
	N = 194	N = 93	N = 39				
RDI, mean (SD) *n*./h	3.71 (3.23)	6.38 (3.92)	25.5 (19.4)	<0.001	0.013	<0.001	<0.001
OA, mean (SD) *n*./h	1.70 (2.24)	2.02 (2.74)	12.6 (15.9)	<0.001	0.907	<0.001	<0.001
H, mean (SD) *n*./h	0.87 (1.24)	0.57 (0.70)	9.29 (7.50)	<0.001	0.670	<0.001	<0.001
OA + H, mean (SD) *n*./h	1.29 (1.43)	1.30 (1.47)	10.9 (8.24)	<0.001	1.000	<0.001	<0.001
CA, mean (SD) *n*./h	0.91 (0.71)	3.53 (1.91)	2.39 (2.74)	<0.001	<0.001	<0.001	<0.001
MA, mean (SD) *n*./h	0.22 (0.63)	0.28 (0.34)	1.37 (2.95)	<0.001	0.886	<0.001	<0.001
ODI	2.26 (2.71)	2.39 (2.24)	21.1 (19.0)	<0.001	0.988	<0.001	<0.001
Minimum SpO_2_, mean (SD) %	87.1 (10.5)	87.4 (7.94)	78.6 (12.5)	<0.001	0.967	<0.001	<0.001
Mean SpO_2_, mean (SD) %	97.1 (1.08)	97.1 (1.79)	94.7 (6.50)	<0.001	0.995	<0.001	<0.001
Time SpO_2_ < 90% eTST, mean (SD)	0.48 (2.29)	1.02 (4.61)	5.95 (15.6)	<0.001	0.768	<0.001	<0.001
Snoring, mean (SD) %TST	2.09 (5.00)	1.68 (4.73)	4.34 (9.72)	0.044	0.839	0.040	0.064
OSAS severity, *n*. (%)				<0.001	<0.001	<0.001	<0.001
Mild	152 (78.4)	39 (41.9)	0 (0.00)				
Moderate	33 (17.0)	42 (45.2)	4 (10.3)				
Severe	9 (4.64)	12 (12.9)	35 (89.7)				
Position							
Supine Sleep Time, mean (SD) %	51.9 (23.9)	48.6 (24.1)	50.0 (27.6)	0.567	0.547	0.954	0.900
Supine OA, mean (SD) *n*./h	2.19 (3.28)	3.07 (5.47)	17.0 (21.8)	<0.001	0.688	<0.001	<0.001
Not supine OA, mean (SD) *n*./h	1.21 (1.79)	1.49 (2.07)	9.72 (14.0)	<0.001	0.900	<0.001	<0.001
Supine H, mean (SD) *n*./h	1.11 (1.83)	0.67 (0.98)	11.0 (10.3)	<0.001	0.641	<0.001	<0.001
Not supine H, mean (SD) *n*./h	0.54 (0.93)	0.46 (0.71)	7.93 (7.22)	<0.001	0.973	<0.001	<0.001
Supine CA, mean (SD) *n*./h	0.98 (1.09)	4.37 (7.96)	3.63 (9.61)	<0.001	<0.001	0.757	0.016
Not supine CA, mean (SD) *n*./h	0.74 (0.61)	3.89 (1.77)	2.51 (2.75)	<0.001	<0.001	<0.001	<0.00
Supine MA, mean (SD) *n*./h	0.25 (0.68)	0.43 (0.75)	1.83 (4.15)	<0.001	0.624	<0.001	<0.001
Not supine MA, mean (SD) *n*./h	0.16 (0.33)	0.22 (0.30)	1.21 (2.55)	<0.001	0.882	<0.001	<0.001

Abbreviations: BMI, Body Mass Index; M, males; SD, Standard Deviation; TST, Total Sleep Time. CA, central apnea; SD, standard deviation; h, hours; H, hypopnea; MA, mixed apnea; *n*./h, number of events/hour; OA, obstructive apnea; ODI, Oxygen Desaturation Index; OSAS, Obstructive Sleep Apnea Syndrome; RDI, Respiratory Disturbance Index; eTST, estimated Total Sleep Time.

## Data Availability

Not applicable.

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
