# Peer review of "Cluster Analysis of Home Polygraphic Recordings in Symptomatic Habitually-Snoring Children: A Precision Medicine Perspective"

_jcm, 2022, doi:10.3390/jcm11195960_

Round 1
Reviewer 1 Report
I have read with interest this article and I would like to congratulate to the authors for this very interesting work. The manuscript presents experimental results about a cluster analysis of home polygraphy measurements in habitually-snoring children. The results are clinically relevant.
I have some recommendations / questions:
- Did the children have any other types of sleep disorders, such as insomnia, parasomnia, delayed sleep phase syndrome etc.? Were there children in the study, who were referred to the clinic because of suspected sleep disorders (please see above: insomnia, parasomnia etc.)?
- Authors should include power calculation in the Methods section.
- English language and style minor spell check is required.
The scientifically sound and methodological principles of the abstract and all paper are appropriate. The text is well structured.
Finally, I recommend the manuscript for revision.
Author Response
Did the children have any other types of sleep disorders, such as insomnia, parasomnia, delayed sleep phase syndrome etc.? Were there children in the study, who were referred to the clinic because of suspected sleep disorders (please see above: insomnia, parasomnia etc.)?
Thank you for your interesting question. Our patients were enrolled because snoring. Together with the instrument, the patients / parents were given a diary for the night, which they filled out and returned. When declared “insomnia, parasomnia, delayed sleep phase syndrome etc” were excluded from the study (line 101) and suggested a neuropsichiatric consultation. However, the parents of our subject studied, to our knowledge, did not declare sleep disorders.
Authors should include power calculation in the Methods section.
We thank the Reviewer for the comment. Since the power calculation has been performed after the cluster identification, we included the information regarding the power calculation in the discussion section (lines 257-261).
English language and style minor spell check is required.
We spell checked as you suggested.
The scientifically sound and methodological principles of the abstract and all paper are appropriate. The text is well structured. Finally, I recommend the manuscript for revision.
We thank for your support.
Reviewer 2 Report
It is nice to do anthropometric cluster analysis, and it showed that in contrary to adult OSA patients, lower BMI means more severe OSA.
However, there is an important anthropometric data missing: Tonsil stage. As we know, the biggest difference between children and adults is that the main reason of child OSA is hypertrophic tonsils and adenoids. Examine adenoid may not be easy, but tonsils are easy to check. Without tonsil grade, other anthropometric data may not be easy to be explained. For example, fat children may have osa without big tonsils,and slim children who snore may have more severe OSA because of big tonsils and fail to thrive. It would be much better to include tonsil grade into cluster analysis.
Moreover, without knowing the underlying mechanism and cause of OSA in children, this kind of grouping will contain more diversified characteristics, not only very hard to be explained in clinical situations, but also provide no clue for future treatment selections, and even cause treatment delay. For example, according to previous literature, tonsillectomy, adenoidectomy and nasal surgery to treat allergic rhinitis have highest success rate in treating pediatric OSA. Those pediatric patients with big tonsils, big adenoids and nasal obstruction may have diversified BMI and AHI severity, and children may also present daily fluctuations in sleep apnea severity. It would be better using tonsil grade,adenoid grade and allergic rhinitis severity as cluster classifier.
The reference 10 have more reasonable cluster results by including tonsil hypertrophy and adenoid hypertrophy. Although that is not a good classification, better using tonsil grade or adenoid grade, that shows how important to include, or at least control tonsil grade when doing cluster analysis.
337 Home sleep test may not be necessary a limitation because patients are more freely movable at home and spend more time in non-supine position, especially in severe OSA patients.(Sensors (Basel). 2021 Dec 3;21(23):8097.)Therefore, home sleep test may allow more POSA data to be investigated in this study, especially for children.
Author Response
It is nice to do anthropometric cluster analysis, and it showed that in contrary to adult OSA patients, lower BMI means more severe OSA.
We thank the reviewer for his/her comment.
However, there is an important anthropometric data missing: Tonsil stage. As we know, the biggest difference between children and adults is that the main reason of child OSA is hypertrophic tonsils and adenoids. Examine adenoid may not be easy, but tonsils are easy to check. Without tonsil grade, other anthropometric data may not be easy to be explained. For example, fat children may have osa without big tonsils,and slim children who snore may have more severe OSA because of big tonsils and fail to thrive. It would be much better to include tonsil grade into cluster analysis.
Moreover, without knowing the underlying mechanism and cause of OSA in children, this kind of grouping will contain more diversified characteristics, not only very hard to be explained in clinical situations, but also provide no clue for future treatment selections, and even cause treatment delay. For example, according to previous literature, tonsillectomy, adenoidectomy and nasal surgery to treat allergic rhinitis have highest success rate in treating pediatric OSA. Those pediatric patients with big tonsils, big adenoids and nasal obstruction may have diversified BMI and AHI severity, and children may also present daily fluctuations in sleep apnea severity. It would be better using tonsil grade,adenoid grade and allergic rhinitis severity as cluster classifier. The reference 10 have more reasonable cluster results by including tonsil hypertrophy and adenoid hypertrophy. Although that is not a good classification, better using tonsil grade or adenoid grade, that shows how important to include, or at least control tonsil grade when doing cluster analysis.
Thank you for your comment.
The classification of tonsils size as a valuable variable to be included in the cluster analysis may be interesting. However, in the study by Wu et al (Reference 10) there was no statistical significance in the incidence of tonsils (p = 0.583) or adenoid hypertrophy (p = 0.787) between clusters, although there were statistically significant differences in AHI and ODI. Therefore, we specified this result by Wu et al in the discussion section. For this reason, we decided not to take account of the dimensional classification of the tonsils in the cluster analysis (lines 354-359). In order to exclude possible “interference” including variables, according also with literature, we did not consider tonsil stage.
Home sleep test may not be necessary a limitation because patients are more freely movable at home and spend more time in non-supine position, especially in severe OSA patients.(Sensors (Basel). 2021 Dec 3;21(23):8097.)Therefore, home sleep test may allow more POSA data to be investigated in this study, especially for children.
Thanks. Agree, the home sleep test was not listed as a research limitation (line 345).
Reviewer 3 Report
Novel way of looking at children with sleep disordered breathing and trying to come up with clusters or groups beyond the current system of classification of severity. However, there are issues that need review:
1. Clarity of number of patients included is needed. There were 470 children; 209 "charts were excluded: - that leaves 261; but 326 were included on the study - not clear about the #s.
2. Method of clustering used has overlaps in RDI: Mean RDI for Cluster 1 3.71+/-3.23; Cluster 2 6.38+/-3.43; Cluster 3 is clearly 25.5+/- 19.4.
3. At the end of clustering method used, n in Cluster 3 is 39 vs 93 in Cluster 2 and 194 in cluster 1. While methodology determined the number in each cluster, emphasizing differences between clusters based in disparate numbers in each group may be overstating the results e.g. lower BMI in cluster 3(with more severe apnea) when compared to others.
4. Other limitations include retrospective study based on home testing but authors recognized that and mentioned that.
5. These issues should be addressed especially in conclusions.
Author Response
Novel way of looking at children with sleep disordered breathing and trying to come up with clusters or groups beyond the current system of classification of severity. However, there are issues that need review:
Clarity of number of patients included is needed. There were 470 children; 209 "charts were excluded: - that leaves 261; but 326 were included on the study - not clear about the #s.
Thank you for reporting the mistakes. The correct numbers are specified in the figure [Some children had more than one exclusion condition at the same time: registration of < 7 hours (n.25) and / or age < 2 years (n.65) and / or genetic diseases (n.30) and / or neurological diseases (n.9)].
Method of clustering used has overlaps in RDI: Mean RDI for Cluster 1 3.71+/-3.23; Cluster 2 6.38+/-3.43; Cluster 3 is clearly 25.5+/- 19.4.
We thank the Reviewer for the comment. However, even if some overlap can occur, it was demonstrated that time taken in cluster head election and space complexity of overlapping of cluster is much better in K-Medoids than K-Means [Arora, Preeti, and Shipra Varshney. "Analysis of k-means and k-medoids algorithm for big data." Procedia Computer Science 78 (2016): 507-512.].
At the end of clustering method used, n in Cluster 3 is 39 vs 93 in Cluster 2 and 194 in cluster 1. While methodology determined the number in each cluster, emphasizing differences between clusters based in disparate numbers in each group may be overstating the results e.g. lower BMI in cluster 3(with more severe apnea) when compared to others.
We thank the Reviewer for the comment. Researchers are frequently faced with the problem of unequal cluster sizes in the population. However, to use K-medoids, instead of K-means may be preferable, even when the data did not contain outliers or noise variables, demonstrated the most accurate classification of individuals overall [Finch, W. H.. A Comparison of Clustering Methods When Group Sizes Are Unequal, Outliers Are Present, and in the Presence of Noise Variables. American Educational Association Multiple Linear Regression Special Interest Group, 2019, doi:10.31523/glmj.045001.003.]
Other limitations include retrospective study based on home testing but authors recognized that and mentioned that.
We thank the Reviewer for the comment.
These issues should be addressed especially in conclusions.
We included comments on limitations in the last paragraph of discussion section.
Round 2
Reviewer 2 Report
The reference 10 is a China study. There are obvious racial differences between your study and their’s. I hope to see the researcher to include tonsil stage in cluster analysis, even if the result is negative, it will make this study more clinically relevant. Otherwise, this study is not complete.
Author Response
Dear reviewer,
We are very sorry, but we cannot pursue what you have asked.Unfortunately, when we designed the study according to the reference10 (Wu Y, et al. Int J Pediatr Otorhinolaryngol. 2019), we did notconsider the tonsils grade.The other two reviewers accepted all our corrections and they do not have anyother questions.It is impossible to add tonsils grade; we need to go back to the Ethical Committee, which will require at least six months. Therefore, your interesting suggestion could be considered in a future study. We hope that you will understand that our position is in line with the recentliterature.Regards